# Adsorption of Vi Capsular Antigen of *Salmonella* Typhi in Chitosan–Poly (Methacrylic Acid) Nanoparticles

**DOI:** 10.3390/polym11071226

**Published:** 2019-07-23

**Authors:** Raimundo Lopes da Silva, Jaqueline Rodrigues da Silva, Anivaldo Pereira Duarte Júnior, Patrícia Santana Barbosa Marinho, Lourivaldo Silva Santos, Francisco Martins Teixeira, José Otávio Carréra Silva Júnior, Roseane Maria Ribeiro Costa

**Affiliations:** 1Laboratory of Pharmaceutical Nanotechnology, College of Pharmacy, Federal University of Pará, Pará 66075-110, Brazil; 2Nanobiotechnology Laboratory, Department of Genetics and Morphology, Institute of Biological Sciences, University of Brasília, Brasília 70910-900, Brazil; 3Department of Chemistry, Institute of Natural Sciences, Federal University of Pará, Pará 66075-110, Brazil; 4College of Pharmacy, Campus Macaé, Federal University of Rio de Janeiro, Macaé, Rio de Janeiro 27930-560, Brazil; 5Laboratory R & D Pharmaceutical and Cosmetic, College of Pharmacy, Federal University of Pará, Pará 66075-110, Brazil

**Keywords:** nanoparticles, chitosan, adsorption, Vi antigen, *Salmonella* Typhi

## Abstract

The development of a nanoparticulate system for the carrier antigen is now an important tool in the vaccination process, since a smaller number of doses is necessary for effective immunization. Thus, in this work a nanoparticulate system using polymers of chitosan and poly (methacrylic acid) (CS–PMAA) to adsorb the Vi antigen of *Salmonella* Typhi was developed. CS–PMAA nanoparticles with different proportions of chitosan and poly (methacrylic acid) were obtained and reached sizes from 123.9 ± 2.48 to 234.9 ± 2.66 nm, and spherical shapes were seen in transmission microscopy. At pH 7.2, the nanoparticles had a cationic surface charge that contributed to the adsorption of the Vi antigen. Qualitative analyses of the isolated Vi antigen were performed using Fourier-transform infrared spectroscopy, which indicated the presence of all the characteristic bands of the capsular polysaccharide, and nuclear magnetic resonance, which showed signals for the five hydrogens and the N-acetyl and O-acetyl groups which are characteristic of the Vi antigen structure. In the adsorption kinetics study, the Vi capsular antigen, contained in a phosphate buffer solution of pH 7.2, experienced 55% adsorption on the 1–1% (CS–PMAA) nanoparticles. The adsorption kinetics results showed the ability of the nanoparticulate system to adsorb the Vi antigen.

## 1. Introduction

Nanotechnology is increasingly playing a significant role in vaccine development [1]. The need for new controlled release systems has been important in establishing more efficient immunotherapies, allowing for the administration of bioactive molecules with greater safety, minimal side effects and greater permanence in circulation. Nanoparticulate systems consisting of biodegradable polymers attract great attention from researchers compared to other systems, such as liposomes, because of their therapeutic potential and stability in biological fluids [2,3,4].

In the development of controlled release systems, such as micro and nanoparticles, polymers are commonly used as a matrix for antigen loading. Chitosan was chosen because it presented several biological activities, such as biocompatibility, biodegradability, non-toxicity and anti-inflammatory, anti-microbial and anti-hypertensive properties [5,6,7]. It also has mucoadhesive properties, which have been widely reported in the literature [8,9].

Chitosan is a natural polysaccharide derived from chitin deacetylation and can be obtained industrially through the hydrolysis of the aminoacetyl groups of chitin, which is found mainly in shellfish carapaces, as well as insects, mollusks and fungal cell walls [10,11]. Chitosan nanoparticles can be obtained by combining chitosan with other polymers, such as poly (methacrylic acid). The introduction of a second component in the formulation increases the versatility of the system in terms of association and its susceptibility to interact with biological surfaces [12]. 

Capsular polysaccharides are important surface structures for the virulence of many Gram-negative bacteria. *Salmonella* Typhi is the etiologic agent of typhoid fever and produces a capsular polysaccharide known as the “Vi antigen” [13].

The Vi antigen is anti-phagocytic and an important virulence factor of *Salmonella* Typhi because it avoids phagocytosis of the bacterium by polymorphonuclears (PMNs) and increases resistance against the oxidative process after phagocytosis. It can also be found in *Salmonella* Paratyphi C, *Salmonella Dublin* and *Citrobacter ballerup* [14]. 

The use of the Vi antigen in parenteral vaccines began very successfully, providing good immunity and producing about 90% of the anti-Vi antibody, but presented concomitant adverse reactions such as pain and local hardening in 20 to 30% of the individuals who needed more than one dose of the vaccine, in addition to rare cases of fever. One of the ways to try to avoid adverse reactions caused by the antigen is the development of a controlled release system consisting of a polymer, such as chitosan, which can reduce the number of doses while obtaining the desired immune response for a longer period of time [15]. 

These controlled release systems are a versatile platform for the delivery of antigens to the immune system because their surfaces can be easily modified to target specific receptors and thus release their encapsulated charge accurately and sustainably. In addition, such particulate systems are capable of protecting antigens and their adjuvants from degradation before they reach the target cells [16,17].

In this context, the aim of the present work is to develop a nanoparticulate system using chitosan and poly (methacrylic acid) as carrier vehicle for the Vi capsular antigen of *Salmonella* Typhi as a new alternative of interest in the field of immunization.

## 2. Materials and Methods

### 2.1. Materials

The *Salmonella* Typhi Ty-2 (ATCC 19430) strains were graciously donated by the Instituto Nacional de Controle de Qualidade em Saúde (INCQS). Pure Vi antigen was obtained from Bio-Rad (Hercules, CA, US) Lot OL 2154. Reagents such as tris (hidroxymethyl) aminomethane (Acs reagent, ≥99.8%, Sigma-Aldrich, St. Louis, MO, USA), hexadecyltrimethylammonium bromide (Lot 089K01611, Sigma-Aldrich) and sodium azide (≥99.5%, Lot STBB7140V, Sigma-Aldrich) were obtained commercially. Desoxyribonuclease I, ribonuclease A and pronase type XIV, were acquired from Sigma-Aldrich. The chitosan with a deacetylation degree of 94% and the 99% methacrylic acid used were supplied by Sigma-Aldrich.

### 2.2. Preparation of Chitosan and Poly (Methacrylic Acid) Nanoparticles (CS–PMAA)

The CS–PMAA nanoparticles were prepared through template polymerization in a methacrylic acid solution containing chitosan according to methodology adapted from Moura, Aouada and Mattoso [18]. Chitosan at mass proportions of 0.5, 0.8 and 1% was dissolved in three solutions of poly (methacrylic acid) (0.5, 0.8 and 1%) (*w*/*v*). Nine formulations were prepared and each one was maintained with magnetic stirring for 12 h. After complete solubilization of the chitosan in the methacrylic acid solution, the formulation was heated until reaching a temperature of 70 °C. For nanoparticle formation, 0.2 mmol potassium persulfate (K_2_S_2_O_8_) was added under stirring for 1 h. The preparation was then cooled in an ice bath and subsequently lyophilized.

### 2.3. Characterization of CS–PMAA

#### 2.3.1. Determination of Hydrodynamic Diameter and Zeta Potential

The particle size and zeta potential of the nanoparticles were determined by photon correlation spectroscopy, using a Zetasizer Nano-ZS90^®^ (Malvern, Worcestershire, UK). Samples were diluted in phosphate buffered in pH 7.2, vortexed and kept for 5 min in an ultrasonic bath. They were then diluted again in PBS (1:100) prior to reading. The whole procedure was performed in triplicate.

#### 2.3.2. Transmission Electron Microscopy (TEM) of Nanoparticles

The samples were sonicated for 2 min for better particle dispersion and to avoid possible particle agglomeration. 10 μL of the CS–PMAA suspension was spread on a carbon-coated copper grid, covered with a carbon film and the excess of water was removed with filter paper. Then, 2% phosphotungstic acid (PTA) was added as a contrast. The homogeneity and morphology of the nanoparticles were evaluated using a transmission electron microscope (ZEISS^®^ EM 900) (Jena, Germany).

### 2.4. Isolation of the Vi Antigen from Salmonella Typhi Ty-2

The antigen isolation procedure was adapted from the technique outlined by Wong and Feeley [19], involving two extraction steps which use several reagents and centrifugations. Briefly, the bacteria were obtained through growth in cell culture bottles, where part were UV-killed and another through treatment with acetone. The cells were suspended in 10 mL of saline solution containing 0.1% sodium azide, shaken for 30 min at 35 °C and centrifuged (20,273× *g*) for 30 min. The supernatant was then treated with 500 µl of desoxyribonuclease (50 µg/100 ml) and ribonuclease (500 µg/100 ml) and incubated in a shaker for 6 h at 37 °C. After that, 50 µl of pronase (0.5 mg/100 ml) was added and incubation was maintained for another 12 h. After that, NaCl was added to final concentration of 5% and the sample was cooled to 2 °C. Then, 2 mL of precooled ethanol was added and the sample was cooled overnight. Afterwards, the sample was centrifuged (20,273× *g*) for 30 min at 2 °C and the pellet was collected. 

Three extracts were made using this sediment by adding a 60% ethanol-saline solution and shaking at 37 °C for 24 h. The extracts were then centrifuged (20,273× *g*) for 30 min at 28 °C to obtain the supernatant. An equal volume of ethanol was added to the supernatant, which was followed by another period of centrifugation (20,273× *g*) for 30 min at 2 °C to obtain sediment. Then, 0.85% saline solution and hexadecyltrimethylammonium bromide were added to a final concentration of 0.1% (*w*/*v*) and centrifuged (10,400× *g*) for 15 min to obtain sediment. A solution of 1 M KCl was added and the sample was filtered through a sintered glass filter. Then, 4 mL of ethanol was added dropwise and the sample was centrifuged (10,400× *g*) for 20 min to obtain sediment. 

At the end of this procedure, sediment was obtained and was resuspended in saline solution. Ethanol was then added dropwise to the concentration of the Vi antigen, thus creating a suspension of the antigen. Two suspensions were obtained, the first from the UV-killed bacteria, which is called Sample 1, and the second from the acetone-killed bacteria, which is called Sample 2. 

### 2.5. Characterization of Vi Antigen

#### 2.5.1. Fourier-Transform Infrared (FTIR) Spectroscopy 

The infrared spectra were obtained using a NICOLET 1179 spectrometer in the wavelength range of 4000 to 400 cm^−1^ for 128 sweeps, with a resolution of 2 cm^−1^. The analysis was performed using both lyophilized samples and samples mixed with potassium bromide (KBr) and then compressed under high pressure to form small pills. Pure lyophilized Vi antigen from SIGMA was used as a standard. 

#### 2.5.2. Nuclear Magnetic Resonance (^1^H NMR) 

For the analysis of hydrogen (^1^H) using an NMR spectrometer (Varian, model Mercury Plus BB 300 MHZ) (Mundelein, IL, USA), 20 mg of each sample was solubilized in deuterium oxide (D_2_O). 

### 2.6. Adsorption Kinetics of the Vi Antigen in CS–PMAA

The kinetics of samples created from 150 mg of 1–1% CS–PMAA nanoparticles added to 30 mL of 1 mg/mL antigen solution in phosphate buffered medium at pH 7.2 at room temperature were analyzed at time intervals of 6, 24 and 48 h. After each interval of 6, 24 and 48, a tube of sample was retired and centrifuged at 4000 rpm for 10 min and the supernatant was collected for subsequent quantification of the non-adsorbed antigen. Quantification of the Vi antigen was performed indirectly using acridine orange, according to an adaptation of the Stone and Szu technique [20], by UV-visible spectrophotometry over the range of 530 to 430 nm (UV Spectrophotometer—Shimadzu 1800^®^) (Kyoto, Japan). 

## 3. Results and Discussion

### 3.1. Characterization of CS–PMAA

#### 3.1.1. Hydrodynamic Diameter and Zeta Potential of CS–PMAA

The particle sizes ranged from 123.9 ± 2.48 to 234.9 ± 2.66 nm (Table 1), which is in agreement with what has been reported in the literature, which is usually between 50 and 300 nm [21]. According to these results, the diameter of the nanoparticles was dependent on the amount of chitosan mass used in the synthesis of the nanoparticles. Thus, it was found that in the formulations with chitosan masses of 0.8 and 1% the particle diameters formed were smaller. The 1–1% chitosan and poly (methacrylic acid) (CS–PMAA) sample presented the smallest nanoparticles (123.9 nm) in relation to all of the other samples. Similar results were observed by Moura, Aouada and Mattoso [18], who prepared CS–PMAA nanoparticles at concentrations of 0.2, 0.8 and 1.0% in a 0.5% methacrylic acid solution, among which the nanoparticles with chitosan masses of 1% presented the smallest diameters.

All nanoparticles prepared had positive charges at pH 7.2, with zeta potential in the range from 2.2 ± 0.5 to 9.8 ± 0.24 mV. This result was important for the adsorption of the antigen, which is negatively charged (−40 mV) at the same pH.

In the literature, the importance of the pH of the medium in relation to chitosan nanoparticles has been reported. At an acidic pH, these nanoparticles have a positive charge, whereas in a basic medium they have a negative charge. The positive charge of the nanoparticles observed in this work is related to the amino group present in the chitosan structure [18,22].

A high zeta potential is important since it creates good physicochemical stability of the colloidal suspension due to high repulsive forces, which tend to prevent aggregation due to collisions of adjacent nanoparticles [23]. 

#### 3.1.2. TEM 

In relation to the morphology of the nanoparticles, similarities were observed between the particles in the various formulations. The nanoparticles showed a regular spherical shape, with size variations occurring within the individual formulations, as well as between formulations (Figure 1). There were few clusters of nanoparticles and quite a few scattered particles.

The dispersion of the nanoparticles was facilitated by the acidic medium in which the nanoparticles were found in these formulations. In an acidic medium the NH_3_^+^ groups of chitosan are protonated, causing repulsion between the particles. The literature has suggested that nanoparticles which are stabilized by electrostatic repulsion experience less aggregation [24]. 

### 3.2. Characterization of Vi antigen

#### 3.2.1. FTIR Spectroscopy

The FTIR spectrum of Sample 1 shown in Figure 2 shows peaks at 617 cm^−1^, corresponding to the vibration of the pyranose ring; at 1101 cm^−1^, due to C–O–C stretching; and between 1650 and 1540 cm^−1^, corresponding to the N-acetyl group. There two peaks, one between 1604 and 1550 cm^−1^ related to the carboxylate anion and a peak at 1734 cm^−1^ from the O-acetyl group. In Sample 2 (Figure 2), the Vi antigen has a peak at 619 cm^−1^, which corresponds to the pyranose ring. There are absorption bands at 1100 cm^−1^ and between 1650 and 1540 cm^−1^, related to C–O–C stretching and the N-acetyl group, respectively. Furthermore, there to peaks related to the O-acetyl group at 1732 cm^−1^, and the carboxylate anion were detected between 1604 and 1550 cm^−1^.

The infrared spectra from Samples 1 and 2 have bands that are related to the chemical composition of the Vi antigen from *Salmonella* Typhi, as has been described in literature. The behavior of the spectra from Samples 1 and 2 shows the presence of all groups of characteristics present in the Vi antigen structure. This demonstrates that the use of UV or acetone to kill the *Salmonella* Typhi cells or the extraction process did not cause a loss of Vi antigen groups.

It is also important to verify the presence of O-acetyl and N-acetyl groups in the Vi antigen structure, since several studies have demonstrated that these groups are related to the immunochemistry of the antigen in *S.* Typhi Ty-2 [25,26].

In both samples the peak for the anion carboxylate group, which is present in the polysaccharide structure, was intense. This highlights the importance of the presence of this group, due to the possibility of creating immunity through the production of antibodies specific to these groups. Furthermore, studies have shown that the anion carboxylate group present in the Vi antigen can form bonds with positively charged groups, thus being useful in vaccine development that uses a controlled liberation system through linkages to positively charged particles, therefore demonstrating the importance of maintaining this group [27].

Santos et al. [28] also described the presence of carboxylic acid between 1629 and 1618 cm^−1^ in a polysaccharide obtained from fractions of *Campomanesia Xanthocarpa Berg.* Analysis using infrared spectroscopy has been used in the identification and characterization of chemical groups related to superficial polysaccharides present in the cell walls of microorganisms. Silva et al. [29] stated that analysis in the infrared region is important for the identification of specific structural characteristics of fungal glucans (polysaccharides that form an external cap around the mycelium). 

#### 3.2.2. Nuclear Magnetic Resonance (^1^H NMR)

Figure 3 shows the ^1^H NMR spectra for Samples 1 and 2. Both samples presented five signals in the 4.63–4.78 ppm range which correspond to the five carbons present on the polysaccharide ring that is present in the structure of pure Vi antigen. The signals corresponding to the N-acetyl and O-acetyl groups were intense in Samples 1 and 2 and were found at 3.29 and 3.28 ppm, respectively. The signal related to the hydrogen H-2 of the polysaccharide ring coincided with the signal for water (4.70 ppm), since pre-saturation of the residual water was not performed in this study. This was also observed by Lemercinier et al. [30], in the NMR analysis of pure Vi capsular antigen. The objective of the pre-saturation technique is to suppress the residual water signal through the application of a long pulse at low energy at the frequency of the solvent.

Characterization of the Vi antigen by NMR was performed by Martínez et al. [31], which analyzed the purity of three commercial lots of Vi antigen using as default the Vi antigen spectrum, which was obtained from the National Institute for Biological Standards and Control (NIBSC, United Kingdom), reporting similarities between their samples and the Vi antigen standard. Kothari et al. [32] characterized the Vi antigen using NMR in the search for a new polysaccharide purification method.

### 3.3. Kinetics of Adsorption 

The adsorption kinetics of the Vi antigen in nanoparticles can be seen in Figure 4. The amount of antigen present in the solution is inversely proportional to the amount of antigen that has been adsorbed on the particles. The rate of adsorption of the antigen on the particles was 55% within 24 h and then remained constant until 48 h. The Vi antigen concentration decreased significantly in the 10 h, with a decline near 24 h, where after this period the fall was interrupted, becoming constant. The decrease in the Vi antigen concentration in the first 10 h can be attributed to the availability of a large number of available sites on the nanoparticles, which decreases until it becomes saturated.

The loading of a drug into a nanoparticulate system can be done by two methods, during preparation (embedding) or after particle formation (incubation). The latter is known as the adsorption method, which is a process that occurs between the absorber, which is usually a solid, and the adsorbate, which may be liquid [33].

Based on this idea, the CS–PMAA nanoparticles represent the solid phase of the system and the Vi antigen represents the liquid phase. The nanoparticles chosen for adsorption were of the following composition: 1% chitosan and 1% methacrylic acid, 123.9 nm in diameter with a surface charge of +7.45 mV. The 1–1% (CS–PMAA) nanoparticles were chosen because they contain a greater chitosan mass, which increases the amount of NH_3_^+^ groups and allows for a higher amount of negatively charged Vi antigen binding. In the literature, it has been proven that the higher the chitosan concentration in CS–PMAA nanoparticles, the better the adsorption efficiency [22]. In addition, the smaller the nanoparticle size, the greater the adsorption efficiency.

In an adsorption system, a balance between the adsorbate and adsorbent occurs when there is no variation in the concentration of adsorbate in the medium under study. In this work we observed a balance after approximately 24 h, that is, there was no adsorption of the Vi antigen on the CS–PMAA nanoparticles after this period, indicating the saturation of the binding sites. 

In the zeta potential study carried out in this work, it was observed that at pH 7.2 the surface charges of all the CS–PMAA nanoparticles were positive, whereas the Vi antigen had a negative charge of −40 mV. Thus, it is considered that adsorption occurs through a chemical interaction based on electrostatic attraction between the positive surface of the 1–1% (CS–PMAA) nanoparticles and the negative charge of the Vi antigen in a medium of pH 7.2. 

Studies have indicated that the negativity of the Vi antigen occurs at the carboxyl group, whereas the positivity of the nanoparticles occurs at the amine group. Therefore, it is probable that a chemical bond forms between these groups in the process of the adsorption kinetics [25].

The results of this study demonstrate that 1–1% (CS–PMAA) nanoparticles presented more than 50% of adsorption of the Vi antigen, which can be explained due to the higher quantity of chitosan mass (1%) used in their formulation. This led to more protonated amino sites of chitosan and the smallest diameter of nanoparticles in relation to the other formulations studied, thus providing a larger contact surface.

## 4. Conclusion

The adsorption kinetics were 55% in the first 24 h for 1–1% CS–PMAA nanoparticles in a solution containing the Vi antigen at a concentration of 1 mg/ml at pH 7.2. Thus, the results obtained in this study show that it is possible to obtain a nanoparticulate system of chitosan modified by poly (methacrylic acid) to adsorb Vi capsular antigen from *Salmonella enterica* serotype Typhi. This is fundamental for future studies related to obtaining a vaccine against typhoid fever, given the efficiency of the CS–PMAA nanoparticles and the immunogenicity of the antigen.

## Figures and Tables

**Figure 1 polymers-11-01226-f001:**
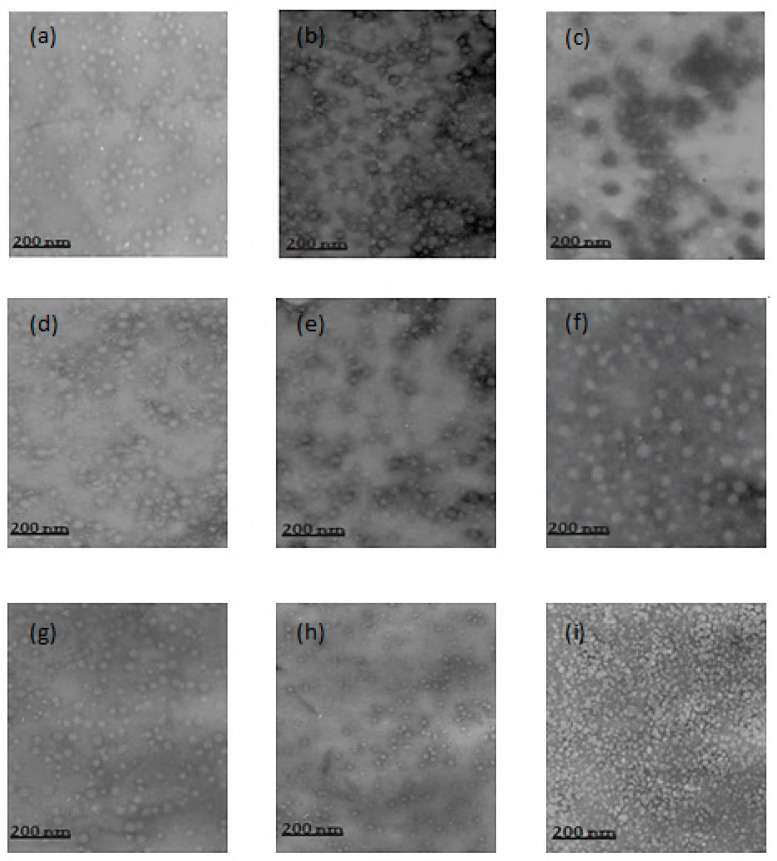
Transmission electron microscopy image of (**a**) 0.5–0.5%, (**b**) 0.8–0.5%, (**c**) 1.0–0.5%, (**d**) 0.5–0.8%, (**e**) 0.8–0.8%, (**f**) 1.0–0.8%, (**g**) 0.5–1.0%, (**h**) 0.8–1.0%, (**i**) 1.0–1.0% CS–PMAA nanoparticles.

**Figure 2 polymers-11-01226-f002:**
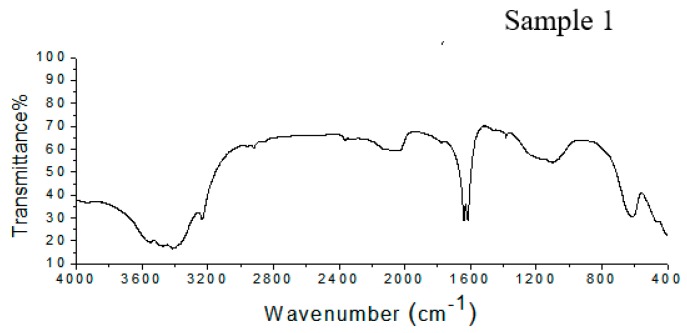
FTIR spectra of Samples 1 and 2, showing the presence of all of the characteristic peaks of the Vi antigen.

**Figure 3 polymers-11-01226-f003:**
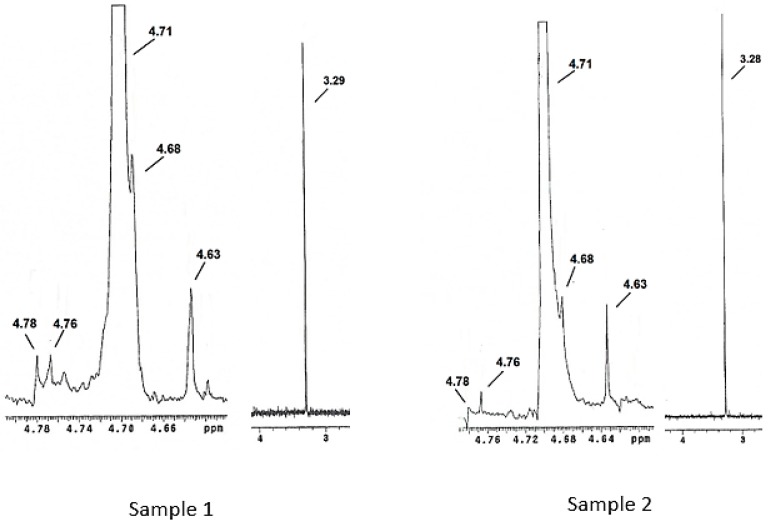
NMR spectra for Samples 1 and 2, showing the five signals corresponding to the chemical structure of the Vi antigen.

**Figure 4 polymers-11-01226-f004:**
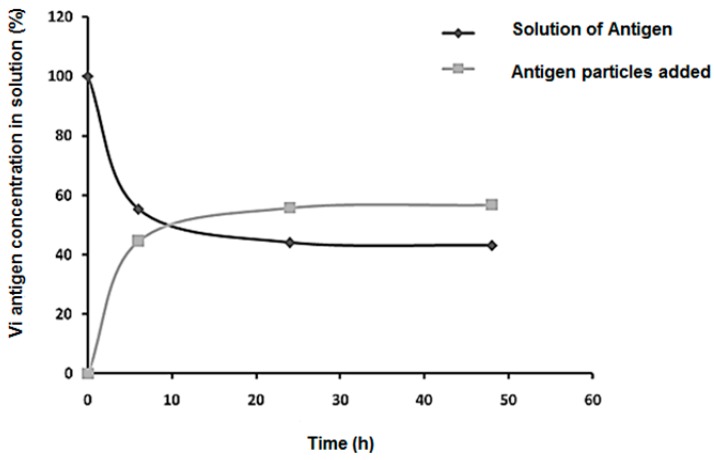
Graph of the kinetics of adsorption of the Vi antigen in a solution of 1–1% (CS–PMAA) nanoparticles at room temperature.

**Table 1 polymers-11-01226-t001:** Hydrodynamic diameter and zeta potentials of chitosan and poly (methacrylic acid) (CS–PMAA) nanoparticles.

CS–PMAA Nanoparticles*	Particle Size (nm)	Zeta Potential (mV)
0.5–0.5% 0.5–0.8% 0.5–1%	213.9 ± 3.29 222.7 ± 10.5 193.2 ± 1.69	4.8 ± 0.55 6.79 ± 0.20 5.55 ± 0.13
0.8–0.5% 0.8–0.8% 0.8–1%	234.9 ± 2.66 173.7 ± 8.27 131.4 ± 0.29	2.2 ± 0.50 6.69 ± 0.23 6.34 ± 0.10
1–0.5% 1–0.8% 1–1%	153.9 ± 0.70 206.8 ± 2.57 123.9 ± 2.48	9.87 ± 0.24 2.65 ± 0.37 7.45 ± 0.15

* Chitosan and poly (methacrylic acid) (CS–PMAA) nanoparticles with different concentrations of chitosan and poly (methacrylic acid). All presented values are the mean ± standard deviation.

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
