# Peer review of "Adsorption of Vi Capsular Antigen of Salmonella Typhi in Chitosan–Poly (Methacrylic Acid) Nanoparticles"

_polymers, 2019, doi:10.3390/polym11071226_

Reviewer 1 Report

MS reports on chitosan-poly(methacrylic acid) nanomaterials for drug delivery applications. In oprinciple some interesting results are provided but the MS needs major revision to improve its clarity and impact.
- Introduction is very short and it needs revision considering recent findings on nanocomposites based on chitosan and their applications (see for instance: Polymers 2019, 11(5), 818; https://doi.org/10.3390/polym11050818; ACS Appl. Nano Mater. https://doi.org/10.1021/acsanm.9b00487; Polymers 2019, 11(5), 839; https://doi.org/10.3390/polym11050839; New J. Chem. 2018, 42, 8384–8390. https://doi.org/10.1039/C8NJ01161C).
- Discussion sometimes is hard to follow.
- Abstract. Abbreviations should be defined one time only. For instance in NMR characterization a lot of technical details are provided but the results are quickly discussed.
- In experimental part details on the used chemicals should be added (purity, Mw for the polymers,…)
- DLS results. “Particles Size” should be changed in hydrodynamic diameter or radius to be clear on the meaning.
- Standard deviations for “size” and Zpotential should be added in the table for a proper comparison.
- Figs 2 and 3 needs to be revised and submitted in a better resolution.

Author Response

Follows attached the Response to Reviewer 1

Reviewer 2 Report

The paper presents the results of adsorption of Vi capsular antigen of Salmonella Typhi in chitosan–poly (methacrylic acid) nanoparticles. 

The paper is interesting, but  needs major revision as follows:

1) Nine CS-PMAA nanoparticles of different sizes, depending on the concentrations, were obtained, but only 1% - 1% (CS-PMAA) were added to antigen solution. It will be more interesting to study the effect of four concentrations if not all of them, on adsorption of Vi capsular antigen of Salmonella Typhi, considering the Conclusion section paragraph where is stipulated :

"it is possible to obtain a nanoparticulate system of chitosan modified by poly (methacrylic acid) which can adsorb Vi capsular antigen from Salmonella enterica serotype Typhi which is fundamental for future studies related to obtaining vaccine against typhoid fever".

2) taking into account the first observation, all figures from 2 to 4 must contain at least four concentrations which subsequently will be series of sample 1 UV-killed bacteria  and sample 2  acetone-killed bacteria (in total at least eight samples to be studied)

3) Comments on the adsorption mechanism must be added

4) A conclusion concerning the concentration-particle size influence in the adsorption of Vi capsular antigen of Salmonella Typhi will be very interesting

Author Response

Follows attached the Response to Reviewer 2 

Reviewer 3 Report

The Abstract and Introduction needs to be improved.

Is not clear and justified the use of poly (methacrylic acid) since this polymers not biodegradable and its use for vaccination is questionable.

Correct throughout the manuscript the numbers in which the comma should be replaced by points.

The data presented is from how many samples? You need to present the n and also the standard deviation.

What is sample 1 and 2 presented FTIR analysis?

FTIR analysis of empty nanoparticles and nanoparticles with adsorbed antigen is lacking.

In order to be able to say that this system will gradually release the antigen it is necessary to present results of the release assays.

Author Response

Follows attached the response to Reviewer 3

Round  2

Reviewer 1 Report

I have a concern on point 7.

Point 7: Figs 2 and 3 needs to be revised and submitted in a better resolution

 Response 7: We do not own more the original image of Figure 2 to improve resolution. In relation to the image 3 we only have it printed, being possible only its scanning, being difficult the improvement in the resolution.

I'm wondering if the authors have the right to publish the figure if they do not own the image.

I leave the decision to the editor. Anyway the figures are not clear and labels not easy to read so probably can be moved to supporting information.

Author Response

Follows in attached the reviewer response

Reviewer 2 Report

the paper has been clearly improved by adding  comments and explanation and can be published in present form

Author Response

Nothing to declare

Reviewer 3 Report

Data needs to be corrected concerning the significant numbers of the mean and the standard deviation values.

Author Response

(The authors gave the same response as above.)
